# Tensor decomposition of multi-dimensional splicing events across multiple tissues to identify splicing-mediated risk genes associated with complex traits

Yan Yan[1☯], Rui Chen[2,3☯], Hakmook Kang[1], Yuting Tan[2,3], Anshul Tiwari[2,3], Siyuan Ma[1], Zhexing Wen[4], Xue Zhong[2,5], Bingshan Li[2,3]*

1 Department of Biostatistics, Vanderbilt University, Nashville, Tennessee, United States of America,
2 Vanderbilt Genetics Institute, Nashville, Tennessee, United States of America, 3 Department of Molecular Physiology and Biophysics, Vanderbilt University, Nashville, Tennessee, United States of America, 4 Department of Psychiatry and Behavioral Sciences, Emory University, Atlanta, Georgia, United States of America, 5 Department of Medicine, Division of Genetic Medicine, Vanderbilt University Medical Center, Nashville, Tennessee, United States of America

☯ These authors contributed equally to this work.
* bingshan.li@vanderbilt.edu

## Abstract

Identifying risk genes associated with complex traits remains challenging. Integrating gene expression data with Genome-Wide Association Study (GWAS) through Transcriptome-Wide Association Study (TWAS) methods has discovered candidate risk genes for various complex traits. Splicing, which explains a comparable heritability of complex traits as gene expression, is under-explored due to its multidimensionality. To leverage multiple splicing events in a gene and shared splicing across tissues, we develop Multi-tissue Splicing Gene (MTSG), which employs tensor decomposition and sparse Canonical Correlation Analysis (sCCA) to extract meaningful information from high-dimensional multiple splicing events across multiple tissues. We build MTSG models using GTEx data and apply them to GWAS summary statistics of Alzheimer's disease (AD) (111,326 cases and 677,663 controls) and schizophrenia (SCZ) (36,989 cases and 113,075 controls). We identify 174 and 497 significant splicing-mediated risk genes for AD and SCZ, respectively, at Bonferroni correction. For AD, our results demonstrate significant enrichment of AD related pathways and identify additional AD risk genes not detected in the single-tissue analysis, while preserving most top genes identified in the brain frontal cortex. Consistently, for SCZ, genes identified by our brain-wide MTSG model, built from a cluster of 13 brain tissues, exhibit stronger enrichment in SCZ-relevant genes and MTSG identifies unique SCZ risk genes compared to single-tissue models. These results showcase that our MTSG models capture distinctive splicing events across tissues, which might be overlooked when using single tissue alone. Our MTSG models can be applied to other complex traits to help identify splicing-mediated disease risk genes.

**Data availability statement:** The open-source code and companion files are in https://github.com/cgg-L/MTSG/tree/main GTEx: https://gtexportal.org/home/downloads/adult-gtex/bulk_tissue_expression GWAS summary statistics of Alzheimer's disease: https://www.ebi.ac.uk/gwas/studies/GCST90027158 GWAS summary statistics of schizophrenia: https://walters.psycm.cf.ac.uk/.

**Funding:** The work was partially supported by the National Institutes of Health (NIH) https://www.nih.gov [R01AG069900 to Y.Y., R.C., Y.T., A.T., Z.W., X.Z., B.L., and R01AG065611 to Y.Y., R.C., Y.T., A.T., Z.W., X.Z., B.L.]. The funder had no role in the study design, data collection and analysis, decision to publish, or preparation of the manuscript.

**Competing interests:** The authors have declared that no competing interests exist.

## Author summary

Splicing is a critical biological process that generates diversity in protein production by rearranging segments of genetic material. It plays an essential role in human health and disease but remains challenging to study due to its complexity, especially when analyzing splicing across multiple tissues. Most existing methods examine splicing in a single tissue at a time, potentially overlooking important cross-tissue patterns. To address this, we developed a novel computational framework called Multi-tissue Splicing Gene (MTSG). By using advanced mathematical techniques, such as tensor decomposition and sparse canonical correlation analysis, MTSG integrates splicing data from multiple tissues to identify genetic risk factors for complex diseases. Applying MTSG to large genetic datasets for Alzheimer's disease and schizophrenia, we identified numerous genes associated with these conditions that were missed by single-tissue analyses. Additionally, the enrichment of these genes in disease-related pathways supports the reliability and relevance of the gene sets identified by our framework. MTSG not only improves our understanding of how splicing contributes to disease but also provides a flexible tool that can be applied to other complex traits. This work paves the way for more comprehensive investigations into the genetic regulation of human diseases, offering new opportunities for therapeutic discovery.

## Introduction

Genome-Wide Association Studies (GWAS) have made tremendous success in identifying the risk loci; however, as most variants are located in the non-coding region of the genome and exhibit their effects through regulation of the target genes, risk genes that mediate the effect of variants on the trait cannot be easily determined. Transcriptome-Wide Association Studies (TWAS) [1,2] are able to nominate candidate risk genes by integrating genotype, expression, and phenotype via expression quantitative trait loci (eQTL) analysis. Using models built on large scale data, e.g. GTEx where both expression and genotype data are available, TWAS has been applied to a vast range of published GWAS to infer the expression-trait associations on the gene-level to identify candidate risk genes [2–4], advancing the mechanistic understanding of genetics of complex traits. Nevertheless, gene regulation is complex, and gene expression accounts for only partial heritability. Splicing, which creates different isoforms by retaining different combinations of exons, has been shown to account comparable heritability as gene expression [5,6]. As an essential step in gene regulation, splicing contributes extensively to protein diversity and has shown to have multiple roles and be critical in cell cycle, adipogenesis, metabolic process, cancer progression and other process [7–10]. While gene expression data has been explored widely, very few studies have used splicing information for risk gene discovery, largely due to the complexity of splicing and the challenges of handling high-dimensional features of splicing.

A statistical framework, named MSG, that jointly models high-dimensional splicing and genotype data in a gene, has been developed to identify genes associated with complex traits [11]. Compared to TWAS methods [4,12,13], MSG identified more risk genes, showing the improved power of joint modeling of multiple splicing events. However, the MSG method is limited to single tissues, and for most complex traits multiple tissues are implicated. Recently TWAS has been extended to joint modeling of multiple tissues, and showed improved power than single tissues in identifying trait associated genes [3,4,12,14]. The success of multiple-tissue TWAS supports that complementary and common regulatory patterns exist across tissues, and we reason that, for splicing, joint modeling of splicing events across multiple tissue is able to similarly boost the statistical power of identifying splicing-mediated risk genes associated with complex traits.

Nevertheless, one challenge for modeling splicing data is its high dimensionality. To tackle this issue, we propose a new computational method, Multi-tissue Splicing Gene (MTSG), a tensor decomposition-based framework to model multidimensional splicing events across tissues. Tensor decomposition, well-established in chemometrics, signal-processing and image analysis, has unique advantages in dimension reduction and latent structure detection [15,16]. Recently, the application of tensor decomposition was extended to genomics studies and showed potential in handling high-dimensional data [17–19]. In brief, MTSG frames the multi-tissue splicing data into a 3-dimensional subject-splicing-tissue tensor, and then decomposes the tensor into three matrices, each containing the loadings of subjects, splicing events, and tissues for each factor. The subject loadings, which indicate the magnitude of latent components across factors within each subject, are then used to extract sparse linear combination of latent components that maximizes the correlation with the genotypes, via sparse canonical correlation analysis (sCCA) [20] (see Methods) to build sparse gene-based splicing predictive models. Association tests are then performed using these models and GWAS summary statistics, yielding results for gene-trait associations.

We built MTSG models using GTEx data [21] (v8), specifically the splicing events inferred by LeafCutter [22], and applied the models to the summary statistics of the GWAS of Alzheimer's disease (AD, 111,326 cases and 677,663 controls) and schizophrenia (SCZ, 36,989 cases and 113,075 controls), and compared the results with those obtained using single tissues alone. We explored the function of identified candidate risk genes, and observed that candidate risk genes identified in our methods showed greater relevance in AD and SCZ, showcasing the advantages of our methods in capturing disease relevant splicing across tissues to boost the power of identifying splicing-mediated risk genes.

## Results

### Method overview

The workflow of the MTSG framework is shown in Fig 1. For each gene, the multi-tissue splicing data can be formatted into a 3-D array $\mathcal{G}$, or in tensor terminology, a third-order tensor, where subjects, splicing events and tissues are on the first, second and third dimension, respectively. Using canonical polyadic decomposition [16], the third-order tensor can be decomposed into 3 matrices, with $Y$ the subject loading matrix (subjects × components), $C$ the splicing loading matrix (splicing × components), and $T$ the tissue loading matrix (tissues × components). Here the components are latent structures that are common across tissues, with contribution of splicing events represented by different loadings in the splicing loading matrix.

Since each subject has a score for each component, and each component contains information of multiple splicing events extracted from multiple tissues, we can use the subject loading matrix $Y$ and the genotype data (matrix $X$) as input to sCCA to build a sparse gene-based splicing-mediated predictive model as described previously [11]. The resulting canonical variables (CVs) maximize the correlation between splicing events and genotypes and represent the splicing components that are genotype regulated. We then perform association analysis using the gene-based predictive models with GWAS summary statistics to identify candidate risk genes. Details are in Methods. Note that MTSG is agnostic to how splicing events are called and therefore applicable to splicing events inferred by other methods.

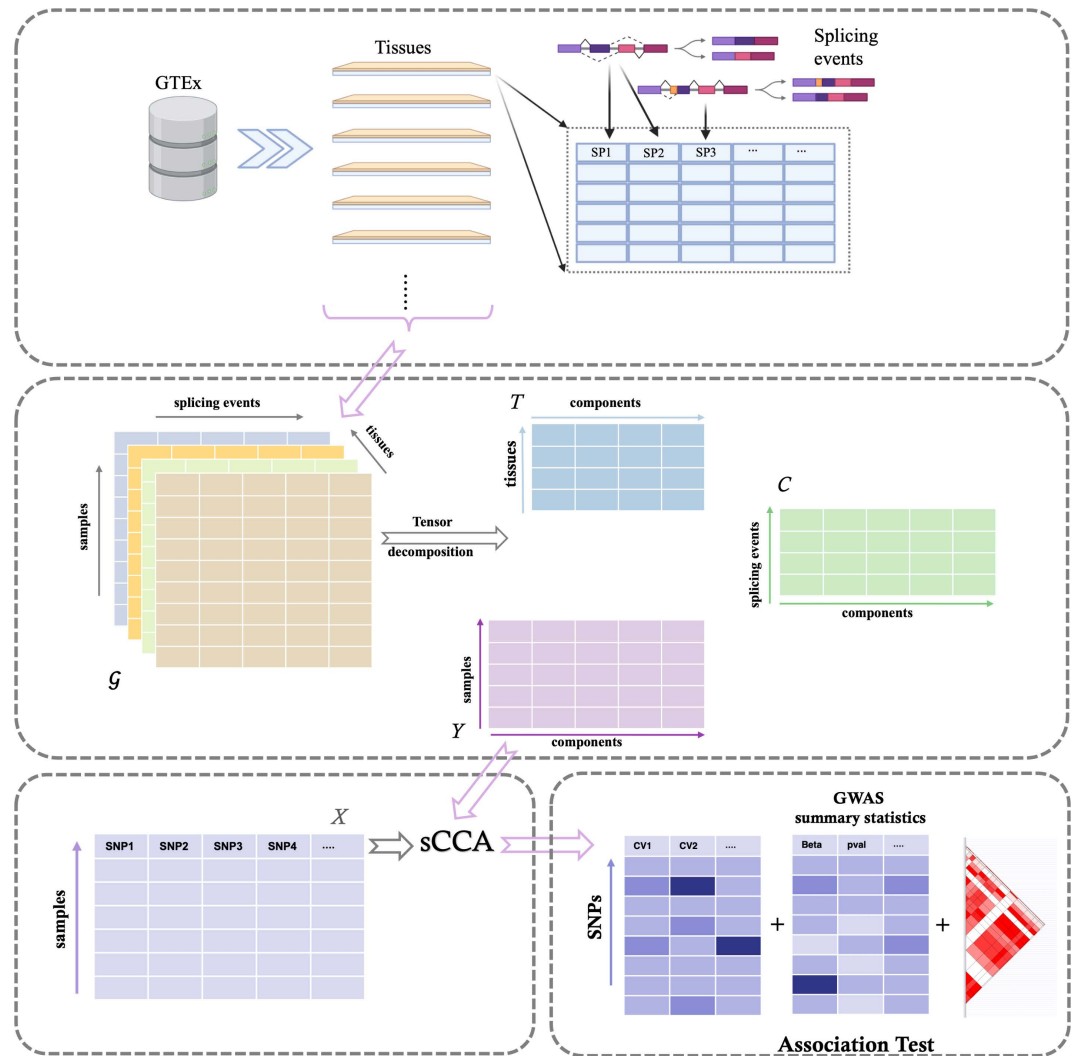

**Fig 1. Overview of the MTSG workflow.** Starting from a third-order tensor $\mathcal{G}$ (samples × splicing events × tissues), MTSG decomposes the tensor into the splicing loading matrix $C$, tissue loading matrix $T$, and subject loading matrix $Y$. Matrix $Y$ contains the scores of each component for all individuals. These scores are then fed into sCCA along with genotype information (Matrix $X$, where SNPs within 1 Mb of the gene transcription starting site are aligned for each individual in Matrix $Y$). The resulting canonical vectors (CVs) are combined with GWAS summary statistics and the LD matrix for association testing between the gene and the trait of interest. LD (Linkage Disequilibrium) matrix is a symmetric matrix that quantifies the correlation between genetic variants (i.e., SNPs). Created with Biorender.com.

## Type I error rates

To estimate the type I error, using GTEx data [21] (v8 release), we simulated the GWAS summary statistics and LD matrix, then combined with the $X$ and $Y$ matrices of genes for a replication of $10^8$ (see Methods). The type I errors of MTSG are well-controlled under various alpha-levels (0.05 at 5%, 0.01 at 1%, $1 \times 10^{-3}$ at 0.1%, $1.02 \times 10^{-4}$ at 0.01%, $9.82 \times 10^{-6}$ at 0.001%, $1.12 \times 10^{-6}$ at 0.0001%).

## Application to AD GWAS summary statistics

We built our MTSG models using 32 tissues of GTEx [21] (see Methods), and applied the models to the GWAS summary statistics of AD [23] (111,326 cases and 677,663 controls) to identify genes associated with the trait. As a comparison, we

also built gene-based splicing models using a single tissue, i.e., brain frontal cortex BA9 in GTEx, which has been used previously for TWAS of AD [13]. In total, MTSG identified 174 significant genes after Bonferroni correction at 5% level (S1 Table for the list of genes), while the single tissue models identified 160, with 88 genes in common, showing increased power of MTSG over the single-tissue models. The Jaccard Index (JI), a similarity metric for comparing two sets, ranges from 0 (no overlap) to 1 (complete identity). Our analysis revealed a JI of 0.358 between the gene sets of MTSG and BA9. Furthermore, we computed JIs between MTSG and all 32 individual tissues, with values ranging from 0.272 to 0.36 (S1 Fig). This relatively narrow range suggests that the signals detected by MTSG are distributed evenly across the tissues.

We also compared the MTSG with all single-tissue analyses. Our single-tissue analyses identified 416 significant genes after multiple testing correction, with each tissue contributing 79–135 genes. Notably, 175 genes (42%) exhibited strong tissue specificity, being detected in only one or two tissues, likely enriched for false positives, considering that tissues often have shared regulatory mechanisms. In contrast, the majority of genes (111/131) identified by both MTSG and individual tissues showed cross-tissue consistency, supporting our method's ability to capture biologically robust signals through cross-tissue information integration. Importantly, MTSG uniquely detected 47 genes, including established AD risk genes (*APOE*, *BIN1*), demonstrating its ability to capture robust, multi-tissue signals while filtering tissue-specific noise.

**Genes identified by MTSG are more enriched in pathways related to AD pathogenesis.** We then performed the Gene Ontology (GO) enrichment analysis for the two sets of significant genes. As shown in Fig 2, genes identified by MTSG are predominantly enriched in multiple terms of biological processes related to AD pathogenesis, including regulation and negative regulation of amyloid precursor protein catabolic process, metabolic process of amyloid precursor protein and amyloid-$\beta$, regulation of amyloid-$\beta$ clearance, regulation of immune system process and innate immune response. Genes identified using the single brain frontal cortex BA9 tissue are only significant in very few of the amyloid-$\beta$ protein related terms with much less significant *P*-value than multiple-tissue identified genes. Besides the amyloid-$\beta$ protein, recently studies have shown that inflammatory response and immune response [24,25] are key factors that contribute to AD pathogenesis. Among the significantly enriched GO terms, MTSG identified multiple terms in immunity and inflammation, e.g., regulation of immune system process, innate immune response, immune system development and antigen processing and presentation, while the single-tissue models are enriched in none. These results strongly demonstrate that MTSG is able to effectively capture splicing information of AD risk genes, while models based on single tissue of BA9 perform poorly.

We further evaluated genes uniquely identified by either multi-tissue models or single-tissue models. Genes unique to MTSG showed significant enrichment in terms related to immunity process (S2 Fig), while those identified uniquely by single-tissue models of BA9 did not show meaningful enrichment of AD relevant GO terms.

**MTSG is better powered to identify AD risk genes.** We further examined the top ranked genes to evaluate the strength of the association and the relevance to AD pathogenesis. *CLASRP*, reported to be AD associated in GWAS study [26] and predicted to be involved in RNA splicing and mRNA processing, is the top gene in the association tests for both multi-tissue models and single-tissue models of BA9, with a stronger *P*-value for the multi-tissue models (*P*-value $= 4.415 \times 10^{-92}$ and $1.216 \times 10^{-74}$ for MTSG and single-tissue models respectively). Following that we have *PPP1R37* (MTSG *P*-value: $7.082 \times 10^{-56}$, single tissue *P*-value: $7.921 \times 10^{-26}$) and *MARK4* (MTSG *P*-value: $1.715 \times 10^{-48}$, single tissue *P*-value: $5.388 \times 10^{-23}$), both of which are known AD risk-genes [27]. Other strong AD risk genes on the top list include *MS4A6A* [28,29] (MTSG *P*-value: $9.064 \times 10^{-15}$, single tissue *P*-value: $3.495 \times 10^{-3}$), and *PICALM* [30] (MTSG *P*-value: $1.657 \times 10^{-25}$, single tissue *P*-value: $1.499 \times 10^{-24}$). These results demonstrate that MTSG models are more powered to identify genuine AD risk genes mediated through splicing.

**MTSG identified additional AD risk genes and preserved many of those identified by single-tissue models of BA9.** We observed that some well-known AD risk genes are only uniquely identified by MTSG but missed by the single-tissue models. *APOE* (*P*-value: $2.961 \times 10^{-21}$) and *BIN1* (*P*-value: $6.340 \times 10^{-34}$) are well-known and extensively studied AD risk genes [23]; however, they were not identified in single-tissue models. Additional AD risk genes uniquely identified by

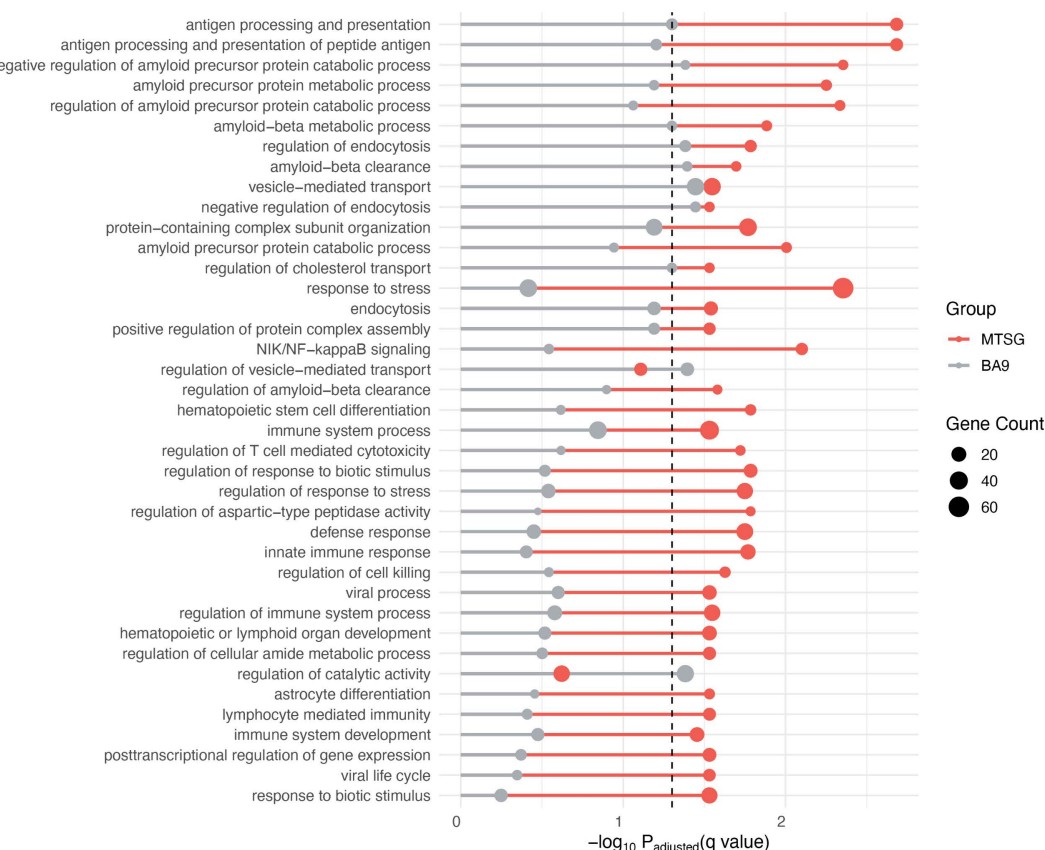

**Fig 2. Comparison of Gene Ontology enrichment of AD risk genes.** Compared to genes identified by brain frontal cortex BA9 alone (grey), genes identified by MTSG (red) show enrichment in much more disease-related pathways. The dash line represents the 0.05 significance cutoff (-log₁₀ 0.05). The size of the circle is proportional to the corresponding gene counts of the enrichment.

MTSG include *BCL3* (*P*-value: $4.191 \times 10^{-25}$), which is associated with genetic linkage with late-onset familial Alzheimer's disease [26,31,32]; *NECTIN2* (*P*-value: $3.539 \times 10^{-29}$), which has strong signals in multiple GWAS and shown to be implicated in the pathology of AD [33]; *HIKESHI* (*P*-value: $1.622 \times 10^{-27}$), which encodes protein Hikeshi, a nuclear import carrier for heat shock protein 70 (HSP70) that inhibits amyloid-$\beta$ oligomerization, enhances amyloid-$\beta$ clearance, restores tau homeostasis, and suppresses neuronal apoptosis [34].

We also explore how the MTSG preserves the signals identified by single tissue. Our analyses demonstrate strong concordance between MTSG and single-tissue results, with approximately 50% overlap in significant genes. Notably, when examining the 147 significant genes identified by MTSG that were testable in BA9, we observed that 81% ranked within the top 300 BA9 genes, and 85% within the top 500. Conversely, the majority of BA9's 134 significant genes (76% and 85%) similarly ranked within the top 300 and 500 genes respectively in our multi-tissue analysis. These results robustly demonstrate that MTSG effectively preserves the many significant genes from single-tissue analyses while enhancing detection power through cross-tissue integration. Following [35], we compiled a list of reported genes associated with AD (S2 Table) and assessed the enrichment of identified genes in this list by Fisher's exact test. Thirty-two genes identified by MTSG (*P*-value: $1.14 \times 10^{-27}$) and 27 genes identified by brain frontal cortex BA9 (*P*-value: $2.83 \times 10^{-27}$) are in the list.

**Biological interpretation of the splicing matrix *C* and tissue matrix *T*.** While the sample matrix *Y* serves for gene-trait association testing, we focus on extracting biological insights from the splicing loading matrix *C* and tissue loading

matrix *T*. By ranking tissues based on their loadings in matrix *T*, we identified whole blood (53 genes), brain cerebellum (50 genes), brain cortex (44 genes), brain nucleus accumbens basal ganglia (43 genes), and liver (41 genes) as the top contributors of significant genes (S3A Fig). This tissue distribution pattern remained largely consistent for genes uniquely identified by MTSG, while the overlapping genes between MTSG and BA9 showed some variation, with pancreas and cultured fibroblasts appearing in the top ranks (S3B Fig). Notably, the prominence of immune-related whole blood and lipid metabolism-enriched liver tissues aligns with known AD pathology [36,37].

At the gene level, we observed distinct tissue and splicing specificity patterns. *TREM2*, a well-characterized AD risk gene, showed its strongest tissue association with spleen (Component 1), corresponding to a specific splicing loading in its matrix *C* (S4A and S4B Fig), and Component 3 exhibited brain cerebellum specificity while capturing two alternative splicing forms. Similar patterns emerged for other AD-associated genes like *APOE* and *TSPAN14*, demonstrating how the decomposed components preserve biologically relevant tissue-specific splicing information (S4C–F Fig). These findings collectively highlight the utility of tensor decomposition not only for association detection but also for uncovering tissue-specific regulatory mechanisms underlying disease.

**Genes with splicing-mediated mechanisms in AD.** We conducted a literature review to confirm that our MTSG approach effectively captures splicing-mediated risk genes in Alzheimer's disease and provides meaningful biological insights. For example, *APOE*, one of the most well-characterized AD risk genes, exhibits functionally consequential alternative splicing in its intron-3 region, which modulates both expression levels and clinical severity in AD [38]. For the *MAPT* gene that encodes tau proteins, alternative splicing produces six major tau isoforms with developmentally regulated expression. Altered expression and aggregation of these isoforms underlie AD [39]. Our analysis of its T matrix demonstrates components showing strong loadings specifically in brain tissues that correspond to multiple distinct splicing isoforms (S5 Fig). Notably, Raj et al.'s study [40] conducted a comprehensive analysis of aberrant splicing patterns incorporating intronic excision levels in Alzheimer's disease (AD), and identified 16 genes associated with AD. Our MTSG results show significant overlap with Raj et al.'s [40] gene set (*P*-value: $= 1 \times 10^{-9}$), including five overlapping genes: *FUS*, *MTCH2*, *PICALM*, *PTK2B*, and *RABEP1*. For example, *PICALM* is involved in both the production and clearance of amyloid-β (Aβ), a principal component of amyloid plaques in AD pathogenesis [41]. Multiple *PICALM* isoforms are expressed in human brain tissue, and overall *PICALM* expression levels - particularly of the major isoform - show significant correlation with AD risk [42]. Our T matrix analysis corroborates these findings, revealing a subset of components with high loadings specifically in brain tissues that correspond to distinct splicing variants (S6 Fig). All others have supportive evidence showing clear implication in AD [43–46].

## Application to SCZ GWAS summary statistics

SCZ is a complex trait with an estimated heritability of 64–81% and prevalence of ~1% globally [47,48]. In contrast to AD, for which causal tissues are less discernible, the etiology of SCZ is primarily linked to brain tissues [49]. However, due to the intricate composition of brain tissues, it is unclear what brain regions and tissues are primarily implicated in SCZ. In fact, it is likely that multiple brain regions and tissues are involved. To fully capture the splicing information carried in brain, we employed a brain tissue cluster approach to build brain-wide MTSG models to identify risk genes associated with SCZ. Specifically, we built our brain-wide MTSG models using a cluster of 13 distinct brain tissues in GTEx (v8, including brain frontal cortex BA9, full list in Methods), and perform association analysis using the GWAS summary statistics of SCZ [50] (36,989 cases and 113,075 controls). Our analysis yielded a total of 497 genes after Bonferroni correction (S3 Table). We conducted a separate analysis focusing solely on the brain frontal cortex BA9, which revealed 476 significant genes. Among them, 292 genes were common to both gene sets, while 205 genes are unique to the brain-wide MTSG, and 184 genes are unique to the analysis of the single-tissue models of BA9. The Jaccard Index between the MTSG and BA9 is 42.9%.

**Gene ontology (GO) analysis.** The GO analysis of the two gene sets showed that they both have a large number of significant terms, which are summarized in Fig 3. The major enriched biological functions are largely the same for the two gene sets, only with different order of relative enrichment, represented by -log$_{10}$ (*Q*-values). Among all terms, interferon-mediated signaling pathway is the leading function for single-tissue models of brain frontal cortex BA9, and amide transport is the top one function for brain-wide MTSG models. Compared with the single-tissue model, brain-wide MTSG identified genes are enriched with greater portion for organelle organization, peptide transport, protein localization and cellular component organization, and slightly smaller for regulation of synaptic plasticity, antigen processing and presentation, and interferon-gamma-mediated signaling pathway.

**Unique genes identified by brain-wide MTSG models are associated with SCZ genetic etiology.** We investigated whether top genes uniquely identified by the brain-wide MTSG models are associated with SCZ, and found that most the top 10 genes are SCZ risk genes. The top gene *ZNF184* (brain-wide MTSG *P*-value: $7.714 \times 10^{-28}$, single tissue *P*-value: 1.0) encodes zinc-binding proteins that regulate transcription and has been reported to be located in a risk loci for SCZ [51]. *TRIM27* (brain-wide MTSG *P*-value: $4.732 \times 10^{-21}$, single tissue *P*-value: 1.0), second on the list, differentially expresses in SCZ and bipolar disorder patients compared to healthy control and is a risk gene identified by GWAS study for SCZ in Japanese and European ancestry [52,53]. The next is *PBX2* (brain-wide MTSG *P*-value: $1.84 \times 10^{-13}$, single tissue *P*-value: 1.0), which was mapped by three variants from a GWAS study for autism spectrum disorder or SCZ [54]. It is also a differentially expressed gene in SCZ patients compared to healthy controls [55]. The fourth on the list, *VPS52* (brain-wide MTSG *P*-value: $8.979 \times 10^{-12}$, single tissue *P*-value: 1.0), implicated in Golgi-associated retrograde protein and endosome-associated recycling protein, has been identified as SCZ candidate marker gene as it is a differentially expressed gene in human BA22 brain samples [56]. These genes would have been missed if using only brain frontal cortex BA9 for analysis, supporting the advantage of modeling the brain tissue cluster in extracting SCZ relevant splicing.

**Genes identified by brain-wide MTSG models enrich more in SCZ relevant gene lists.** Following Wang's work [57], we compiled 9 gene lists involved in SCZ and examined the enrichment of the genes identified by brain-wide MTSG models and single tissue models in these lists. These include gene lists for biological functions that have been shown to be important for SCZ: fragile X mental retardation protein (FMRP) targets, postsynaptic density genes (PSD), genes related to the calcium channel and signaling (CCS), presynaptic active zone (PRAZ), synaptic vesicles (SYV) and presynaptic proteins (PRP). There are multiple FMRP and PSD gene lists, and we only selected the one with best enrichment in their results, i.e., FMRP-Darnell and PSD with 1444 genes. In addition, we also selected one gene set from their autism spectrum disorder (ASD) gene list (included for the shared pathophysiology between psychiatric disorders) - targets of RBFOX1 (RNA binding protein, fox-1 homolog 1), since it is a brain- and muscle-specific splicing factor. As

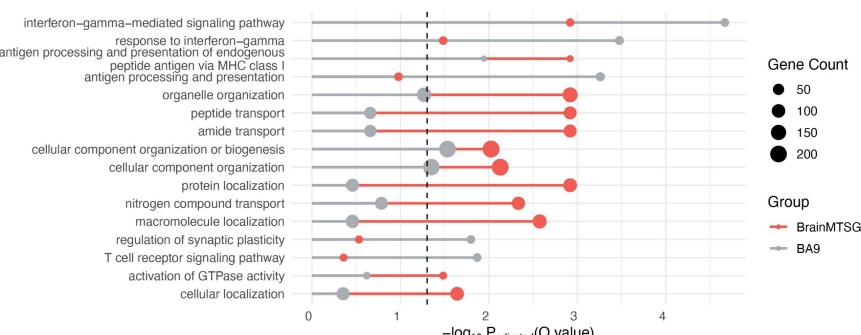

**Fig 3. Comparison of Gene Ontology enrichment of SCZ risk genes.** Risk genes identified by brain-wide MTSG are in red and by brain frontal cortex BA9 are in grey. The dash line represents the 0.05 significance cutoff (-log$_{10}$ 0.05). The size of the circle is proportional to the corresponding gene counts of the enrichment.

shown in Fig 4, genes identified by brain-wide MTSG models exhibit much stronger enrichment in FMRP targets (Darnel), genes related CCS and PSD, and are either higher or comparable in enrichment for most of the functional categories. Notably, genes identified by brain-wide MTSG models enrich more in targets of RBFOX1.

## Discussion

Splicing, impacting an estimated 95% of genes in the human genome [5], assumes a pivotal role in numerous functional pathways and is implicated in disease pathogenesis. It has also become a therapeutic target for certain conditions. Contributing significantly to protein diversity, splicing influences essential cell functions for organismal homeostasis, providing unique information not fully captured by gene expression alone. Exploring splicing in conjunction with expression analysis will deepen our understanding of the underlying regulatory mechanisms. Nevertheless, very few studies have explored splicing, largely due to the challenge of the multi-dimensional nature of splicing data. A few studies model single tissue splicing to identify risk genes associated with complex traits [58–60]; however, such studies carry a heavier multiple-test burden and overlook the inter-tissue correlations.

Here, we develop a new computational framework, MTSG, to model splicing across tissues using tensor decomposition, an approach specifically tailored to address the challenging multidimensional nature of splicing data when analyzed across tissues, as well as to borrow information across tissues to increase power. This method allows us to capture SNP-regulated splicing components across tissues, providing a comprehensive view of the splicing-mediated genetic underpinnings of complex traits. Building MTSG models using GTEx data (v8) and applying the models to AD and SCZ GWAS summary statistics, we showed that the multi-tissue models through tensor decomposition achieved increased power of identifying splicing-mediated disease associated risk genes, compared to single tissue models. Our results not only affirm the potency of our approach but also underscore the wealth of information embedded in splicing events across tissues, offering new opportunities to decipher the intricate genetics governing of complex traits.

Several multi-tissue TWAS models have been developed over the years, e.g., MultiXcan [4], JTI [3], UTMOST [13], and sCCA+ACAT [12], each employing distinct statistical models to integrate data across multiple tissues. Importantly, all these multi-tissue methods have consistently demonstrated enhanced statistical power and efficiency in detecting genetic associations on expression data, highlighting the substantial information gain achieved through the incorporation of multiple tissues in the modeling process. This is consistent with the improved performance of joint modeling splicing events

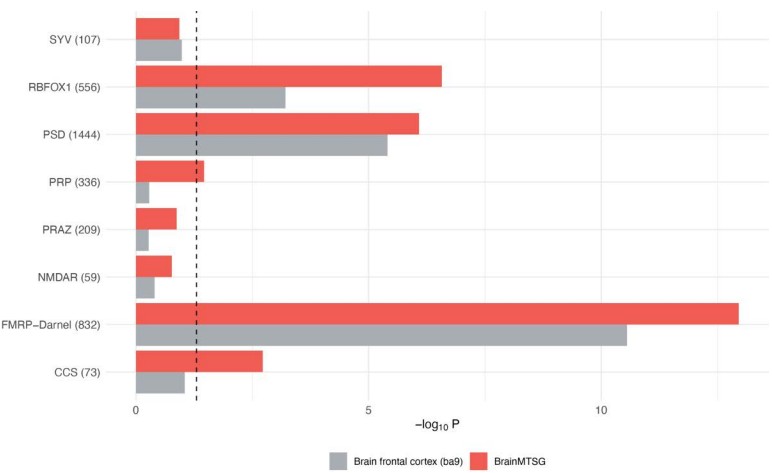

**Fig 4. Enrichment of SCZ risk genes in known gene lists.** The gene lists known to be implicated in SCZ are adapted from [57].Genes identified by brain frontal cortex BA9 are in grey and by brain-wide MTSG are in red. The dash line represents the 0.05 significance cutoff (-log$_{10}$ 0.05).

across multi-tissues in our MTSG framework demonstrated in this study. Rather than treating individual tissues separately and leveraging correlations between them, our method considers the individual-splicing-tissue data as a third-order tensor to maximally preserve the natural structure of the data, providing a nuanced perspective on the interplay between genetic variations and splicing events across diverse tissues. Such modeling strategy enables us to reveal the patterns that would have been missed by separate models due to information loss in brutal dimension collapse. However, it is important to note that the SNP effect captured in our approach, like in other multi-tissue methods, represents an averaged effect across tissues. Consequently, any tissue-specific effects may be masked and may not stand out. While differences in the impact of SNPs across multiple tissues could reveal more detailed regulatory mechanisms, addressing this requires more complex models.

Our results for AD indicated that the clinically relevant tissue may not be the best for risk gene identification and show-cased the power of borrowing information from multiple tissues. The multiple-tissue approach is especially useful when relevant or causal tissue is unknown, which is often the case for complex traits. We acknowledge that if the splicing event is highly tissue-specific or the relevant tissues are clearly determined, then using only the relevant tissues may be a better choice, as demonstrated by using the brain tissue cluster for SCZ. To incorporate prior knowledge and maximize the efficiency, we recommend combing the relevant tissues if they are previously determined and using all available tissues if less is known about which tissue/tissues are causally associated with the trait.

For association testing across multiple tissues, imputation is necessary due to varying sample availability across different tissues. The accuracy of this imputation step may introduce additional variability. We evaluated two widely used imputation packages: MICE (Multivariate Imputation by Chained Equations) [61] and Amelia. Using a subset of complete data with simulated 10% random missingness, we compared the root mean square error (RMSE) between imputed and observed values. Amelia demonstrated superior performance with a lower RMSE (0.18 vs. 0.24 for MICE), indicating better reconstruction accuracy, and was consequently selected for downstream analyses. While other methods (e.g., k-nearest neighbors) may not scale effectively for multi-tissue splicing data or handle mixed data types as robustly, recent advances in deep learning have shown promising results for multi-tissue data imputation [62,63]. Future improvements could incorporate these state-of-the-art deep learning approaches.

Though we only used the subject loading matrix for analysis after the tensor decomposition, the other two matrices, splicing loading matrix and tissue matrix contain information to explore. Specifically, the tissue score indicates the magnitude of each component in each tissue and will be helpful in exploring the across-tissue pattern of components. The splicing loadings will help us learn more about the correlation pattern of the different splicing events. The interpretability of canonical polyadic decomposition used in our framework makes it more competent than other decomposition approaches for application in biological context.

Our framework demonstrates broad applicability for identifying splicing-mediated risk genes across diverse complex traits. Successful applications to HDL [64] (183 genes), IBD [65] (419 genes), MDD [66] (462 genes), and RA [67] (323 genes) have revealed disease-relevant pathways that align with established pathological mechanisms, confirming the generalizability of our approach (S4 Table).

Furthermore, technology advances, e.g., single cell-omics, generate fine-scale high-dimension data across cell types, or multi-omics data across time points, open a door for a broader application of the methods to model such data. The potential of our approach to be extended to higher dimensions presents an opportunity to accommodate the increasingly information-rich complex genomics datasets to dissect genetic underpinning of human diseases.

## Methods

For a given gene, we can construct a third-order tensor $\mathcal{G}$ of size $n \times r \times t$, where $n$, $r$, and $t$ denote number of individuals, number of components, and number of tissues, respectively. Following [19], after decomposition algorithm, three factor matrices will be produced. The decomposed subject loading matrix $Y \in \mathbb{R}^{n \times r}$ has each sample in the row and each

component in the column. The number of components $r$ is set to $p$ (number of splicing events for each gene). The selection of the rank is supported by the following considerations: First, our model is theoretically well-defined given the specified $r$ (which is set to $p$), in the sense that it is uniquely identifiable according to the Kruskal uniqueness condition [68] for tensor factorization. Specifically, in our data, the number of splicing events is often small and always below the number of subjects (median = 8, as shown in S7 Fig). This yields that the sum of the rank of subject, splicing event, and tissue loading matrices is $2p + t$ or $3p$, whichever is the smaller. This will always be larger than $2p + 2$, which, according to Kruskal et al.[68], is sufficient to guarantee the uniqueness of the model's factorization formulation. Second, our approach generates a comprehensive representation of genes' possible splicing event patterns, ready to be utilized in the downstream sCCA analyses. By aligning $r$ with the number of splicing events, we will be able to represent all possible splicing combinations for each gene (in its splicing event loading matrix). The corresponding subject loading matrix will thus provide a comprehensive profile of each subject's splicing configurations, for sCCA to effectively extract predictive patterns between SNPs and splicing components. Third, we conducted additional analyses using two alternative rank thresholds: 0.5 and 1.5 times of $p$, comparing with the results presented here. The results demonstrate that approximately 80% of the genes identified under these alternative thresholds were consistently ranked in the top 3% genes in our primary analysis. These variations in gene identification can be attributed to differences in $P$-value distributions across the threshold settings. In this way, we ensure that our tensor decomposition is theoretically sound, biologically interpretable, computationally feasible, and optimized for downstream analyses. For tensor decomposition, we used `cp` function from R package "rTensor" (version 1.4.8) with default settings.

Next, we use $Y$ and genotype matrix $X \in R^{n \times q}$ (with SNPs within the 1-Mb window around the transcription start sites of the gene, $q$ denotes the number of SNPs) to find the sparse linear combinations of columns of $Y$ and $X$ that are maximally correlated with each other, using sCCA [20] implemented in the R package "PMA". Our objective is to identify $w_1$ and $u_1$ that maximize $w_1^T X^T Y u_1$, while subject to $\|w_1\|_2^2 \leq 1$, $\|u_1\|_2^2 \leq 1$, $\|w_1\|_1 \leq c_1$, $\|u_1\|_1 \leq c_2$. Here $\|\cdot\|_1$ and $\|\cdot\|_2$ denote the $L_1$ and $L_2$ norms, respectively and $c_1$ and $c_2$ are parameters that control the sparsity of $w_1$ and $u_1$, respectively. Following [11], $c_1$ and $c_2$ are chosen by permutation using the "CCA.permute" function in "PMA". With the selected pair of $c_1$ and $c_2$, we derive additional canonical variables (CVs) by continually applying the sCCA algorithm to the updated $X^T Y$ matrix, with the influence of the preceding CVs removed. After repeating the procedure $p-1$ times we obtain $p$ sets of weights $(w_1, u_1)$, $(w_2, u_2)$, ..., $(w_p, u_p)$ and collect $(w_1, ..., w_p)$ into $W$, the $q \times p$ matrix of SNP weights.

The output is a gene-based cross-tissue splicing predictive model. The resulting CVs- and weights are then used to test the association with GWAS summary statistics of trait of interest through Wald test. Let $z$ be the GWAS summary statistics for the association between genotype and the trait of interest. The association between genetically regulated splicing CVs and the trait is given by $W^T z$. Under the null hypothesis of no association, $W^T z$ follows a multivariate normal distribution with mean 0 and covariance $W^T \Sigma W$, where $\Sigma$ is $q \times q$ LD matrix that can be estimated from an external reference panel. Thus, the following statistic follows a $\chi^2$ distribution and can be used to test the association between the gene and the trait.

$$T = z^T W \left( W^T \Sigma W \right)^{-1} W^T z$$

Due to the typical correlation among splicing events in a given gene, the covariance matrix $W^T \Sigma W$ can approach singularity, making its inverse difficult to be estimated reliably. Singular value decomposition (SVD) regulation was used to compute the pseudo-inverse of covariance matrix $\left( W^T \Sigma W \right)^-$ as described previously [11], with small eigenvalues removed (less than 1/30 of the maximum eigenvalue).

We utilized the splicing data from v8 release of GTEx project [21], which provides the multiple-tissue expression and splicing data across 49 tissues of 848 donors. The splicing events were calculated as intron excision ration by LeafCutter [22], which analyzes mapped reads that span exon-exon junctions to detect intron excision events. Note that not all

donors have data available for all tissues. After removing tissues with less than 200 available samples and subjects with less than 13 tissues, we got a final data set from 32 tissues of 581 donors (S5 Table for the list of 32 tissues). To handle missing values in the splicing data, we implemented distinct approaches for two types of missingness. For biologically absent splicing events (where variants are not expressed in certain tissues), missing values were imputed using the minimum observed value across the splicing matrix as a conservative estimate approaching biological zero. For technical missing data resulting from incomplete individual coverage across tissues, we performed statistical imputation using R package "amelia" to account for gaps in sample measurements. This dual strategy ensures appropriate treatment of both biological reality and technical limitations while maintaining data integrity for multi-tissue analyses. Splicing events that are missing in more than 20 tissues were removed from analysis to ensure the quality of imputation. The imputed data were used to format the third-order tensor for each gene.

For SCZ, we used the brain tissue cluster to build the brain-wide MTSG models, as it is known that brain tissues are the most relevant tissues for SCZ [49]. Specifically, we selected samples in GTEx that have data for at least one brain tissue, filtering out those with data for less than 5 tissues, and finally obtained 192 samples with 13 brain tissues (S6 Table for the list of 13 brain tissues). The same imputation methods were used as above to fill in missing splicing data. Genes with less than two splicing events were excluded. Bonferroni correction was applied at 0.05 level for the genome-wide significance using only the testable genes.

To evaluate the type I error of MTSG, we used the GTEx data to construct the $X$ and $Y$ matrix and simulated the GWAS summary statistics. For the external linkage disequilibrium (LD) matrix, we used 5,000 randomly selected sample of European ancestry from BioVU [69], the biobank at Vanderbilt University Medical Center that contains DNA samples from routine clinical testing. For GWAS summary statistics, we first simulated individual level genotype matrix $X_1$ ($n \times q$) and the trait $Y_1$ ($n$) and compute the summary statistics. We set $n = 5000$, $q$ equal to the number of SNPs for a specific gene, and only use the summary statistics for analysis. We simulated rows of $X_1$ independently from multi-variate normal distribution with mean 0, variance 1 and covariance matrix determined by the LD matrix of the SNPs of the gene. Under the null, there is no association between $X_1$ and $Y_1$, so we simulated $Y_1$ from a standard normal distribution. For each testable gene in GTEx, we repeated the above simulation 10,000 times. In total, the simulation ran over $10^8$ times. A $P$-value cutoff of 0.05 was used.

## Supporting information

**S1 Fig. Counts of AD risk genes identified using 32 individual tissues.**
(TIF)

**S2 Fig. GO analysis of genes identified uniquely by MTSG for AD.**
(TIF)

**S3 Fig. Top contributor of tissues of significant genes for AD. By ranking tissues based on their loadings in matrix *T*, we identified which tissues are the top contributors to significant genes.** A: MTSG genes, B: Overlapped genes between MTSG and BA9.
(TIF)

**S4 Fig. Heatmap of *T* and *C* matrices of *TREM* (A,B), *APOE* (C,D) and *TSPAN14* (E,F).** (A,B) *TREM2*, showed its strongest tissue association with spleen (Comp_1), which corresponded to a specific splicing isoform in matrix *C*. Comp_3 exhibited brain cerebellum specificity while capturing two alternative splicing forms. (C,D,E,F) Parallel patterns emerged for other genes *APOE* and *TSPAN14*, demonstrating how the decomposed components preserve biologically relevant tissue-specific splicing information.
(TIF)

**S5 Fig.  *T* and *C* matrices of *PICALM* in AD.** The rectangles show the small group of components (components 2, 11, 16 and 19) with high loading in brain related tissues and in a small group of splicing.
(TIF)

**S6 Fig.  *T* and *C* matrices of *MAPT* in AD.** Components 7 shows very high loading in brain related tissues and several splicing forms.
(TIF)

**S7 Fig.  Distribution of number of splicing events among all testable genes in the genome.**
(TIF)

**S1 Table.  AD risk genes.**
(XLSX)

**S2 Table.  Reported AD risk genes.**
(XLSX)

**S3 Table.  SCZ risk genes.**
(XLSX)

**S4 Table.  Other diseases.**
(XLSX)

**S5 Table.  List of tissues.**
(XLSX)

**S6 Table.  Brain tissues.**
(XLSX)

## Author contributions

**Conceptualization:** Yan Yan, Rui Chen, Bingshan Li.

**Data curation:** Yan Yan, Rui Chen, Yuting Tan, Anshul Tiwari, Xue Zhong.

**Formal analysis:** Yan Yan, Rui Chen.

**Funding acquisition:** Zhexing Wen, Bingshan Li.

**Investigation:** Yan Yan, Rui Chen, Hakmook Kang, Yuting Tan, Anshul Tiwari, Siyuan Ma, Zhexing Wen, Xue Zhong.

**Methodology:** Yan Yan, Rui Chen, Bingshan Li.

**Project administration:** Yan Yan.

**Resources:** Yan Yan, Zhexing Wen, Bingshan Li.

**Software:** Yan Yan, Rui Chen.

**Supervision:** Bingshan Li.

**Validation:** Yan Yan, Rui Chen, Bingshan Li.

**Visualization:** Yan Yan.

**Writing – original draft:** Yan Yan, Bingshan Li.

**Writing – review & editing:** Yan Yan, Rui Chen, Hakmook Kang, Yuting Tan, Anshul Tiwari, Siyuan Ma, Zhexing Wen, Xue Zhong, Bingshan Li.

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
