## [Decision Letter · Decision Letter 0]

Tensor decomposition of multi-dimensional splicing events across multiple tissues to identify splicing-mediated risk genes associated with complex traits

PLOS Computational Biology

Dear Dr. Li,

Thank you for submitting your manuscript to PLOS Computational Biology. After careful consideration, we feel that it has merit but does not fully meet PLOS Computational Biology's publication criteria as it currently stands. Therefore, we invite you to submit a revised version of the manuscript that addresses the points raised during the review process.

Please submit your revised manuscript within 60 days May 26 2025 11:59PM. If you will need more time than this to complete your revisions, please reply to this message or contact the journal office at ploscompbiol@plos.org. Please include the following items when submitting your revised manuscript:

We look forward to receiving your revised manuscript.

Kind regards,

Wei Li, Ph.D.

Academic Editor

PLOS Computational Biology

Shihua Zhang

Section Editor

PLOS Computational Biology

**Journal Requirements:**

At this stage, the following Authors/Authors require contributions: Yan Yan, Rui Chen, Hakmook Kang, Yuting Tan, Anshul Tiwari, Zhexing Wen, Xue Zhong, and Bingshan Li. Please ensure that the full contributions of each author are acknowledged in the "Add/Edit/Remove Authors" section of our submission form.

Potential Copyright Issues:

i) Figure 1. Please confirm whether you drew the images / clip-art within the figure panels by hand. If you did not draw the images, please provide (a) a link to the source of the images or icons and their license / terms of use; or (b) written permission from the copyright holder to publish the images or icons under our CC BY 4.0 license. Alternatively, you may replace the images with open source alternatives. See these open source resources you may use to replace images / clip-art:

**Reviewers' comments:**

Reviewer's Responses to Questions

Reviewer #1: The Multi-Tissue Splicing Gene (MTSG) model, utilizing tensor decomposition and sparse Canonical Correlation Analysis (sCCA), effectively integrates multi-tissue splicing event data to identify splicing-mediated risk genes associated with complex diseases such as Alzheimer's Disease (AD) and Schizophrenia (SCZ). This innovative approach not only demonstrates the potential of multi-tissue models in improving the identification of disease-related genes but also enhances the statistical power compared to single-tissue models. The genes identified by the model are more relevant to the genetic etiology of these diseases, and the multi-tissue analysis captures splicing events that may be overlooked in single-tissue studies, thereby providing valuable insights for personalized medicine. In total, this work makes a valuable contribution by developing the MTSG model, significantly enhancing the identification of splicing-mediated risk genes in complex diseases. Here are some concerns:

1. I am curious about the difference between the results obtained using individual tissue and those obtained by integrating all tissues, because the paper demonstrates the advantage of using multiple tissue data to increase the number of identified genes and the statistical power of the analysis. To quantify this difference, maybe the authors could calculate the Jaccard index for the genes identified across different tissues, or something like that. This would allow you to assess the overlap and consistency between the results from individual tissues versus the multi-tissue analysis. Such an analysis would help in understanding the contribution of different tissues to the disease identification process and further validate the reliability of MTSG model.

2. The authors mention that the number of components in the tensor decomposition was chosen based on the number of splicing events for each gene. It would be helpful to include an analysis to show how different choices of the number of components might impact the results. This analysis could help readers understand the potential influence of parameter choices on the identified risk genes and their biological relevance.

3. The authors impute missing data in the GTEx dataset using the "mice" R package. Have the authors tested different imputation methods, and how might the choice of imputation technique influence the results? It would be useful to include a brief discussion on the potential limitations or biases introduced by imputation, especially when dealing with large-scale multi-tissue data.

4. While the manuscript effectively identifies splicing-mediated risk genes, it would be helpful if the authors could further explore the biological impact of specific splicing events on gene function in the discussion section. For instance, the authors could attempt to explain how certain splicing events might alter the structure or function of the resulting protein products and contribute to disease pathogenesis. Understanding the molecular mechanisms by which these splicing events affect gene function could provide deeper insights into their role in complex traits like AD and SCZ. Additionally, examining whether certain splicing events are associated with changes in protein domains or the formation of isoforms with distinct functional properties would enhance the manuscript's biological interpretation and impact.

5. Small typos in the manuscript:

“After repeating the procedure p-1 times we obtain p sets of weights (w1, w2)…”, it should be “(w1, u1)” in the page 21.

“Recently TWAS has been extended to joint modeling of multiple tissues, and showed improved powe than single tissues in identifying trait associated genes.”, it should be “power” in the page 5.

Reviewer #2: In the manuscript “Tensor decomposition of multi-dimensional splicing events across multiple tissues to identify splicing-mediated risk genes associated with complex traits”, the author described a framework called MTSG, which extracts latent factors in mRNA splicing across multiple contexts (e.g. tissues) by tensor decomposition, followed by CCA analysis with genotype data to build a predictive model. This model was then applied to GWAS summary statistics to identify disease-associated genes that potentially mediate disease etiology through splicing. The authors showed that this method is better powered than canonical single-tissue analysis and could be extended to study other diseases and molecular phenotypes. In general, although the method and data analysis in this manuscript look promising, the authors mostly reported high-level results like the number of significant genes and gene-set enrichment analysis. Therefore, I would like to see more details of the tensor decomposition results and how this method compared to other methods more commonly used to link genes to disease GWAS loci.

Major comments:

1. Tensor decomposition is central to the method proposed in this manuscript. But little detail was given about its performance. Instead, the authors directly jump to sCCA analysis and interpretation of gene association results. I would like to see more about the results from tensor decomposition:

a. Specifically, I’m curious about what matrices C and T look like and whether they could provide meaningful information as well. For example, does matrix T show tissue-specific pattern of the components; does matrix C show which splicing events are driving each topic?

2. The authors compared the MTSG method with single-tissue method, using only one tissue in their comparative analysis (brain frontal cortex BA9). A likely scenario is that some splicing events are tissue-specific, so they might not be found in brain frontal cortex BA9. This might explain why MTSG tend to find more genes than single-tissue method. A fairer comparison would be to run single-tissue analysis in all tissues included in the MTSG model and take the union set of significant genes across all tissues.

3. The authors argue that MTSG finds more genes while preserving most top genes in single-tissue analysis (Abstract). But this is not immediately clear by reading the main text: “In total, MTSG identified 174 significant genes after Bonferroni correction at 5% level (Supplementary Table 3 for the list of genes), while the single tissue models identified 160, with 88 genes in common” (line 156-158) and “We conducted a separate analysis focusing solely on the brain frontal cortex BA9, which revealed 476 significant genes. Among them, 292 genes were common to both datasets…” (line 237-238). These numbers suggest that the overlap between MTSG and single-tissue analysis is only about 50%, meaning that the two methods often find very different genes. Therefore, a better comparison of the two gene lists is needed. Maybe the authors could break down the gene lists by certain criteria to support the notion that MTSG can “preserving most top genes in single-tissue analysis”.

4. For AD analysis, it is unclear where the increased power come from compared with single-tissue analysis. Is it because MTSG shares and aggregates information across tissues, or is it because there’s some tissue-specific splicing? I would like to see more analysis on the genes unique to MTSG in AD analysis. Could some signals come from non-brain tissues (AD GWAS colocalize with immune QTL a lot)? If yes, what's the interpretation? Also, related to interpretation, what is the PSI for the introns in MTSG-unique genes across tissues? Can the T or C matrices be used to help answer these questions?

5. Brain tissues are known to have more complex splicing patterns, which are more likely to benefit from MTSG method. I'm curious if MTSG could be applied to other GWAS summary statistics as a comparison, such as rheumatoid arthritis, crohn's disease, T2D or height, all of which have high-quality public GWAS data. Of course I'm not suggesting the authors should do all of these.

Minor comments:

1. In SCZ analysis, only brain tissues were used. What is the rationale? Please explain if possible.

2. Typo on line 81, should be “improved power”, not “improved powe”.

Reviewer #3: This manuscript explores the identification of splicing-mediated risk genes using multi-tissue splicing data and tensor decomposition. While the approach is methodologically interesting, the study lacks sufficient biological justification and interpretability. My detailed comments are as below.

1. The manuscript argues that splicing information provides a finer resolution for identifying risk genes compared to gene expression. However, it does not provide a clear mechanistic rationale for how splicing, rather than gene expression, is functionally linked to the traits examined.

2. The manuscript claims that multi-tissue splicing analysis adds value by leveraging cross-tissue splicing patterns, yet it fails to provide any evidence supporting this claim. A key advantage of multi-tissue data is the ability to identify tissue-specific splicing patterns or risk genes; however, the authors do not present results demonstrating such patterns. Specifically, the study does not highlight which tissues contribute most to the associations, nor does it compare tissue-specific effects to single-tissue models. Furthermore, the identified risk genes are not shown to exhibit distinct splicing patterns across tissues.

3. The manuscript employs tensor decomposition to extract latent components from multi-tissue splicing data, yet it does not leverage the biologically meaningful information embedded in the splicing event-specific and tissue-specific loadings. These loadings provide critical insights into how splicing variations manifest across tissues and how they contribute to trait associations, but the authors do not utilize them in their downstream analysis. Instead, they rely solely on sample loadings without justifying why splicing- or tissue-specific information was excluded. This omission represents a significant loss of potentially valuable biological insights, which is not adequately explained in the study.

4. The manuscript assumes that missing values in splicing data arise from technical artifacts and require imputation. However, splicing event absence in a particular tissue may represent a true biological zero rather than a technical dropout. The impact of this assumption should be examined, as imputing biologically meaningful zeros could introduce biases into the analysis.

5. Several methodological aspects lack sufficient clarity:

- The LD matrix is mentioned but not defined in the main text or figure captions.

- The definition of p (the number of components) is ambiguous. It appears to refer to the number of splicing events per gene, but this should be explicitly stated.

- Page 19, line 360 contains an incomplete sentence, suggesting that some methodological details may be missing.

**Have the authors made all data and (if applicable) computational code underlying the findings in their manuscript fully available?**

Reviewer #1: Yes

Reviewer #2: Yes

Reviewer #3: Yes

PLOS authors have the option to publish the peer review history of their article (what does this mean? ). If published, this will include your full peer review and any attached files.

**Do you want your identity to be public for this peer review?** For information about this choice, including consent withdrawal, please see our Privacy Policy .

Reviewer #1: No

Reviewer #2: No

Reviewer #3: No

**Figure resubmission:**
---

## [Decision Letter · Decision Letter 1]

Dear Dr. Li,

We are pleased to inform you that your manuscript 'Tensor decomposition of multi-dimensional splicing events across multiple tissues to identify splicing-mediated risk genes associated with complex traits' has been provisionally accepted for publication in PLOS Computational Biology.

Best regards,

Wei Li, Ph.D.

Academic Editor

PLOS Computational Biology

Shihua Zhang

Section Editor

PLOS Computational Biology

Reviewer's Responses to Questions

**Comments to the Authors:**

Reviewer #1: The authors have addressed all my concerns.

Reviewer #2: The revision has successfully addressed my comments and concerns.

Reviewer #3: The authors have addressed most of my previous comments. I would suggest also including a short paragraph in the main text that summarizes the trade-offs between single-slice and multi-slice methods. Much of this is already well explained in the authors' response, and summarizing it in the manuscript would help guide users in choosing the appropriate method based on their study goals and tissue relevance.

**Have the authors made all data and (if applicable) computational code underlying the findings in their manuscript fully available?**

Reviewer #1: None

Reviewer #2: None

Reviewer #3: Yes

PLOS authors have the option to publish the peer review history of their article (what does this mean? ). If published, this will include your full peer review and any attached files.

**Do you want your identity to be public for this peer review?** For information about this choice, including consent withdrawal, please see our Privacy Policy .

Reviewer #1: No

Reviewer #2: No

Reviewer #3: No

---

## [Editor Report · Acceptance letter]

PCOMPBIOL-D-25-00193R1

Tensor decomposition of multi-dimensional splicing events across multiple tissues to identify splicing-mediated risk genes associated with complex traits

Dear Dr Li,

I am pleased to inform you that your manuscript has been formally accepted for publication in PLOS Computational Biology. Your manuscript is now with our production department and you will be notified of the publication date in due course.

With kind regards,

Zsuzsanna Gémesi
